# A *Herpetosiphon llansteffanensis* Strain from Forest Soil Exhibits Biocontrol Activity Against Pear Fire Blight

**DOI:** 10.3390/plants14111564

**Published:** 2025-05-22

**Authors:** Wen Lv, Ruiyue Wang, Wenbo Ji, Benzhong Fu, Ming Luo, Jian Han

**Affiliations:** 1Department of Plant Pathology, College of Agronomy, Xinjiang Agricultural University, Urumqi 830052, China; 15292689320@163.com (W.L.); wang18099318221@163.com (R.W.); 18129292664@163.com (W.J.); benzhongf@yahoo.com (B.F.); luomingxjau@sina.com (M.L.); 2Key Laboratory of Prevention and Control of Invasive Alien Species in Agriculture and Forestry of the North-Western Desert Oasis (Co-Construction by Ministry and Province), Ministry of Agriculture and Rural Affairs, Urumqi 830052, China; 3Key Laboratory of the Pest Monitoring and Safety Control of Crops and Forests of Xinjiang Uygur Autonomous Region, Urumqi 830052, China

**Keywords:** predatory, *Erwinia amylovora*, biological control, antibacterial activity

## Abstract

Fire blight, a devastating bacterial disease caused by *Erwinia amylovora*, has posed significant challenges to apple and pear production for over a century. This study introduces a gliding filamentous bacterium, the strain NSD29, isolated from natural forest soil in Xinjiang, China, as a biological control agent for managing this disease. Comprehensive characterization based on morphological, physiological, biochemical, 16S rRNA gene, and whole-genome analyses identified the strain NSD29 as *Herpetosiphon llansteffanensis*. The train NSD29 demonstrated potent predatory activity against *E. amylovora* in vitro. Its biocontrol efficacy was subsequently evaluated on detached leaves, inflorescences, young fruit, and shoots of fragrant pear under controlled greenhouse conditions. Results indicated that applying *H*. *llansteffanensis* NSD29 significantly inhibited lesion expansion on pear leaves and young fruit, achieving protective efficacies of 75.2% and 72.0%, respectively. Furthermore, pre-treatment spraying with NSD29 effectively reduced the incidence of blossom blight, with a control efficacy of 61.2%. On detached pear shoots, the application of NSD29 fermentation broth suppressed lesion expansion, demonstrating substantial protective (86.8%) and curative (75.6%) efficacies. This research provides the first evidence for the potential of *Herpetosiphon* species in the biological control of plant diseases, highlighting *H. llansteffanensis* NSD29 as a promising candidate for developing strategies to combat fire blight.

## 1. Introduction

Fire blight, caused by the bacterium *Erwinia amylovora*, is one of the most devastating bacterial diseases affecting pome fruit trees within the Rosaceae family [1]. This pathogen impacts economically important species such as fragrant pear (*Pyrus sinkiangensis*), apple, hawthorn, and quince [2], and is known to spread with rapidity in fragrant pear orchards [3]. *E. amylovora* typically initiates infection via blossoms, which serve as primary inocula, facilitating subsequent spread to leaves, young shoots, and developing fruit. Furthermore, pruning wounds represent major infection routes. Once established, the pathogen can persist within the host plant, leading to systemic spread and rapid epidemics that are challenging to control and eradicate, resulting in severe economic losses [4,5,6].

The first reported outbreak of fire blight in China occurred in 2016 within the Yili Prefecture of Xinjiang [7]. By 2017, the disease affected approximately 6700 hectares in Korla City, causing yield reductions in fragrant pear ranging from 30% to 50%. The prevalence and spread of fire blight are alarming, with the disease now confirmed and causing substantial economic damage in at least two Chinese provinces: Xinjiang and Gansu [7].

Current management strategies for fire blight encompass quarantine measures, rigorous pruning and removal of infected plant material, chemical applications, biological control tactics, and the breeding of resistant cultivars [8,9,10,11,12]. However, significant challenges persist. Currently, cultivated fruit tree varieties generally lack sufficient resistance to fire blight. While the removal of diseased branches is crucial for limiting pathogen spread, this practice severely impacts fruit yield. Moreover, the widespread application of chemical pesticides, particularly antibiotics like streptomycin, leads to mounting issues, including the development of pathogen resistance, environmental contamination, and concerns over pesticide residues in fruit [13,14,15,16]. Consequently, fire blight remains inadequately controlled in many regions, and the development of safe, effective, and sustainable control methods remains a critical global challenge.

Research indicates that beneficial microorganisms exhibiting antagonistic activity against plant pathogens, utilized either as whole organisms or as their bioactive compounds, can achieve disease control efficacy comparable to conventional pesticides. Biological control offers distinct advantages, such as high target specificity, a lower risk of resistance development in the pathogen population, enhanced safety profiles, and greater environmental compatibility [17,18]. Scientists are actively exploring diverse microbial habitats, including plant surfaces (epiphytes), internal plant tissues (endophytes), and soil ecosystems, in search of antagonistic microorganisms [15,19]. Several promising candidates have been identified, encompassing bacteria such as *Pseudomonas fluorescens*, *Lactobacillus plantarum*, and *Bacillus amyloliquefaciens*, alongside bacteriophages and predatory myxobacteria [20,21,22,23]. However, current biocontrol research and application predominantly focus on a limited number of well-characterized microbial strains, thereby limiting the diversity of the available biocontrol arsenal against fire blight. Therefore, the discovery of novel microbial antagonists with highly effective biocontrol agents has garnered increasing scientific attention.

The genus *Herpetosiphon* represents the only known predatory group within the phylum Chloroflexota. Its predatory behavior towards host bacteria is analogous to the “wolfpack” strategy employed by myxobacteria: cells initially contact the host via gliding motility, then aggregate around the host cell, and finally secrete a variety of hydrolytic enzymes to lyse and assimilate the host. Furthermore, *Herpetosiphon* can produce novel secondary metabolites [24]. These distinctive biological traits suggest that *Herpetosiphon* holds significant potential for application in the biological control of plant diseases. Nevertheless, despite being a novel biocontrol agent with predatory capabilities, its utilization in managing plant diseases within agricultural settings has not received adequate attention.

A strain of *Herpetosiphon*, designated NSD29, was isolated from forest soil in the Tianshan Grand Canyon, Xinjiang Uygur Autonomous Region, China. Through morphological observation, physiological and biochemical characterization, and molecular identification, the strain was determined to be *H. llansteffanensis*. To assess its biocontrol potential, the predatory activity of the strain NSD29 was evaluated against several plant pathogenic bacteria, including the pear fire blight pathogen, using plate assays. Additionally, the biocontrol efficacy of the strain NSD29 was assessed against pear fire blight on detached pear leaves, inflorescences, young fruits, and tender shoots. The discovery of the strain NSD29 provides a scientific basis for exploring new strategies in the biological control of pear fire blight.

## 2. Results

### 2.1. Characterization of Strain NSD29

The strain NSD29, an isolate obtained during myxobacteria isolation (Figure 1A), exhibited distinctive phenotypic characteristics. On VY/2 agar, colonies were initially moist, developing an orange pigmentation with prolonged incubation. Colony morphology displayed a spreading, “membranous” growth pattern, characterized by the formation of aggregated mounds or flame-like protrusions at the colony margin, interconnected by a network of veins (Figure 1B). Light microscopy revealed cells as unbranched filaments exceeding 100 μm in length, composed of chains of bacilli. With culture maturation, these filaments fragmented into individual cells. Scanning electron microscopy (SEM) confirmed the filamentous morphology of individual cells (Figure 1C). Transmission electron microscopy (TEM) further revealed that cells were arranged in extended chains, delineated by distinct cell walls (Figure 1D).

Comparative biochemical profiling against *Herpetosiphon* type strains (Table 1) demonstrated that the strain NSD29 fermented esculin, arginine, and sucrose and exhibited gelatin liquefaction. In contrast, the strain NSD29 did not utilize glucose, arabinose, rhamnose, or mannitol, highlighting significant metabolic differences from established *Herpetosiphon* species.

### 2.2. 16S rRNA Sequence Analysis

The partial 16S rRNA gene of the strain NSD29 (1232 bp) was amplified and sequenced (GenBank accession number PV550079). BLAST (1.4.0) analysis against *Herpetosiphon* type species revealed the highest sequence similarity (99.2%) to *H. llansteffanensis* CA052B^T^ (GenBank accession number PUBZ01000030), *H. giganteus* DSM 589^T^ (GenBank accession number KY689830), and *H. gulosus* NBRC 112829^T^ (GenBank accession number KY689833). Phylogenetic analysis (Figure 2) further supported this close relationship, clustering the strain NSD29 with *H. llansteffanensis* CA052B^T^.

### 2.3. Whole-Genome Features of the Strain NSD29

The NSD29 genome consists of a 5,776,661 bp circular chromosome and four plasmids (237,880 bp, 122,708 bp, 76,840 bp, and 50,164 bp), with an overall G + C content of 50.89% (Figure 3, Table 2). Notably, each plasmid exhibits a slightly higher G + C content (51.11–52.99%) than the chromosome. Assembly quality metrics include an N_50_ of 16,238 bp and an N_20_ of 40,963 bp (NSD29 GenBank CP188824, SRA accession PRJNA1255817). Compared with the three publicly available *Herpetosiphon* genomes (*H. aurantiacus* DSM785^T^ (CP000875), *H. llansteffanensis* CA052B^T^ (PUBZ01000117), and *H. geysericola* DSM7119^T^ (LGKP01000036)), NSD29′s chromosome is similarly sized but distinguished by its plasmid complement and G + C bias (Table 2).

Gene prediction by GeneMarkS identified 5126 protein-coding sequences and 86 RNA genes. Functional annotation with EggNOG-mapper classified 4033 of these as coding regions’ 3326 proteins and 707 pseudogenes with 82.5% assigned to known functions and the remainder annotated as hypothetical proteins. COG functional category distribution is summarized in Table 3. Prophage analysis (PhiSpy) revealed 30 prophage regions, the largest spanning 97,002 bp. AntiSMASH detected eight secondary-metabolite biosynthetic gene clusters: one PKS, two NRPS, two PKS–NRPS hybrids, two terpene clusters, and one thiopeptide cluster (Table 2).

The average nucleotide identity (ANI) between NSD29 and *H. llansteffanensis* CA052B^T^ was 96.1%, exceeding the 95% species threshold, whereas ANI values with *H. giganteus* DSM 589^T^, *H. gulosus* NBRC 112829^T^, *H. geysericola* DSM 7119^T^, and *H. aurantiacus* DSM 785^T^ ranged from 84.4% to 91.7% (Table 4). Digital DNA–DNA hybridization results corroborated these relationships, confirming NSD29 as *H. llansteffanensis*.

Digital DNA–DNA hybridization (dDDH) analysis further supported the distinctiveness of the strain NSD29, yielding a dDDH value of 66.2% with *H. llansteffanensis* CA052B^T^. Pairwise comparisons with other *Herpetosiphon* type strains resulted in significantly lower dDDH values: 45.3% with *H. giganteus* DSM 589^T^, 28.1% with *H. gulosus* NBRC 112829^T^, 43.2% with *H. geysericola* DSM 7119^T^, and 28.2% with *H. aurantiacus* DSM 785^T^. All these values fall below the 70% threshold generally accepted for species delineation [26] (Table 4).

Morphological and 16S rRNA gene sequence analyses suggested the affiliation of the strain NSD29 to the genus *Herpetosiphon*. Further analysis revealed a 16S rRNA gene sequence similarity of 99.2% between NSD29 and the type strain *H. llansteffanensis* CA052B^T^, with both clustering together in the phylogenetic tree. While physiological and biochemical characteristics also showed the greatest similarity to *H. llansteffanensis* CA052B^T^, the dDDH value of 66.2% is below the species boundary. Additionally, average nucleotide identity (ANI) values obtained from genomic comparisons with *H. llansteffanensis* CA052B^T^ were above the 95% species threshold. Therefore, based on the combined evidence of 16S rRNA phylogeny and high genomic similarity (ANI), the strain NSD29 is attributed to *H. llansteffanensis*.

### 2.4. Predatory Activity of the Strain NSD29 Against Plant Pathogenic Bacteria

The strain NSD29 demonstrated predation upon all five tested plant pathogenic bacteria when co-cultured on TPM medium (Figure 4A). Figure 4B shows a schematic representation of the predation process. Quantification of viable prey cells revealed a significant reduction in colony-forming units (CFUs) after NSD29 exposure. Specifically, *E. amylovora* decreased from 2.6 × 10^9^ CFU to 1.8 × 10^6^ CFU, *Dickeya fangzhongdai* from 1.6 × 10^10^ CFU to 2.2 × 10^7^ CFU, *P. syringae* pv. *syringae* from 1.6 × 10^10^ CFU to 2.8 × 10^8^ CFU, *Acidovorax citrulli* from 1.6 × 10^9^ CFU to 5.5 × 10^7^ CFU, and *Pectobacterium carotovorum* subsp. *carotovorum* from 4.7 × 10^9^ CFU to 1.2 × 10^8^ CFU (Figure 4C). These results indicate the predatory capacity of the strain NSD29 against a range of phytopathogenic bacteria.

### 2.5. Evaluation of the Biocontrol Efficacy on Pear Fire Blight


Foliar application of the strain NSD29 exhibited significant protective effects against pear blossom blight. Five days after treatment, the biocontrol efficacy of NSD29 was 62.0%, compared with 73.0% for benziothiazolinone (Figure 5A,B). To further evaluate its efficacy, the biocontrol assay was conducted on pear leaves, fruits, and branches. Spray application of NSD29 significantly inhibited the expansion rate of lesions in pear leaf veins and reduced lesion length, achieving a protective efficacy of 75.2% 5 days after inoculation, compared with 77.4% for benziothiazolinone (Figure 5C,D). Similarly, NSD29 application suppressed lesion expansion and necrosis on immature pear fruits, demonstrating a protective efficacy of 72.0% 5 days after inoculation, while benziothiazolinone showed 80.0% efficacy (Figure 5E,F). Furthermore, assays using detached pear branches inoculated with *E. amylovora* revealed that NSD29 fermentation broth effectively inhibited lesion development and slowed symptom onset, with lesion areas significantly smaller than the control (Figure 6A,C). The protective and curative efficacies of NSD29 7 days after inoculation were 86.8% and 75.6%, respectively, compared with 88.1% and 84.1% for benziothiazolinone (Figure 6B,D).

## 3. Discussion


Fire blight, caused by the bacterium *E. amylovora*, has long been a notorious bacterial disease affecting apple and pear worldwide, with its recent emergence in northwestern China. The overuse of chemical agents in combating this disease has led to detrimental consequences, including the acceleration of antimicrobial resistance development in pathogen populations and a conflict with the principles of sustainable and environmentally sound agriculture. While several microorganisms have been explored for the biological control of fire blight, gaps remain in the comprehensive understanding and application of effective specific strains.

This study aimed to identify antagonistic bacteria capable of inhibiting *E. amylovora* growth. A gliding filamentous bacterium, *H. llansteffanensis* NSD29, was isolated and characterized, which presents a promising addition to the disease biocontrol arsenal. This discovery highlights the potential of *H. llansteffanensis* NSD29 as a biocontrol agent against this destructive plant bacterial disease, offering an innovative and ecologically responsible approach to its management.

The genus *Herpetosiphon*, first described in 1968 [27], currently comprises five recognized species: *H. aurantiacus*, *H. geysericola*, *H. gulosus*, *H. llansteffanensis*, and *H. giganteus* [25,28,29]. These species have been isolated from diverse environments, including algal slime in freshwater, near hot springs, beach sand, plant residues, stream margins, and paddy field sediments. In this study, the strain NSD29 was successfully isolated from complex natural soil using *E. coli* as prey. Preliminary characterization based on morphology, physiological and biochemical assays, and 16S rRNA gene sequencing confirmed the isolate’s affiliation to the genus *Herpetosiphon*. Further comparative genomic analyses revealed that the physiological, biochemical, and genomic features of NSD29 were most closely related to *H. llansteffanensis* CA052B^T^. The average nucleotide identity (ANI) and digital DNA-DNA hybridization (dDDH) values obtained for the strain NSD29 and *H. llansteffanensis* CA052B^T^ fall below the thresholds for defining a novel species. Consequently, based on this integrated analysis, the strain NSD29 is attributed to *H. llansteffanensis*.

Bacterial secondary metabolites play a direct role in plant disease interactions, potentially influencing gene expression signals during cell colonization [30]. It is well established that many crucial microbial secondary metabolites are synthesized through dual synthase systems involving polyketide synthases (PKS) and nonribosomal peptide synthetases (NRPSs), yielding compounds such as erythromycin, cyclosporine, daptomycin, and penicillin [31,32]. Genome-based analyses have revealed the presence of NRPS and NRPS/PKS gene clusters within *Herpetosiphon* genomes, suggesting the capacity for synthesizing antimicrobial secondary metabolites [28]. Indeed, research has identified novel secondary metabolites produced by *Herpetosiphon*, such as siphonazole from *Herpetosiphon* sp. 060, which exhibits significant cytotoxicity [33]. Additionally, auriculamide, a compound from *H. aurantiacus* 114-95, demonstrates potential as an antimicrobial agent [34]. The genome of *H. llansteffanensis* CA052B^T^ harbors one polyketide synthase (PKS), two nonribosomal peptide synthetases (NRPSs), two PKS/NRPS hybrids, two terpene gene clusters, and one thiopeptide gene cluster. Similarly, the genomes of *H. aurantiacus* DSM 785^T^ and *H. geysericola* DSM 7119^T^ contain comparable sets of secondary metabolite biosynthesis gene clusters [25]. Notably, the genomic analysis of the *H. llansteffanensis* strain NSD29 also revealed the presence of one polyketide synthase (PKS), two nonribosomal peptide synthetases (NRPS), two PKS/NRPS hybrids, two terpene gene clusters, and one thiopeptide gene cluster. However, the specific roles of these secondary metabolite biosynthesis clusters during the predation of plant pathogenic bacteria warrant further investigation.

The genus *Herpetosiphon*, currently the only known predatory member within the phylum Chloroflexota, exhibits predatory behavior in all described species except the three earliest documented (*H. cohaerens*, *H. nigricans*, and *H. persicus*), where predation has not been explicitly reported. The predatory nature of *Herpetosiphon* was initially observed by Lewin in *H. geysericola* [35], although a detailed study was not conducted. Quinn et al. [36] systematically investigated the predatory activity of six *Herpetosiphon* strains isolated from freshwater environments, demonstrating varying degrees of predation against a broad range of Gram-negative and Gram-positive hosts. Notably, *Shigella sonnei* NCTC 8230, *Serratia marcescens* UQM 169, and *Ralstonia solanacearum* UQM 1367 were completely lysed, while *E. coli*, *B. subtilis*, and *Micrococcus luteus* were not susceptible to predation [36]. Livingstone et al. [25] reported that *H. llansteffanensis* CA052B^T^ exhibited predatory activity against *E. coli*, *Klebsiella pneumoniae*, *Proteus mirabilis*, *Staphylococcus aureus*, *S. epidermidis*, *S. saprophyticus*, *Enterococcus faecalis*, *B. subtilis*, and *Candida albicans*. In this study, it was observed that the strain NSD29 displayed predatory activity against five Gram-negative plant pathogenic bacteria. However, the predatory efficiency of the strain NSD29 varied significantly depending on the prey species. For instance, in plate co-culture assays, the viable cell counts of *E. amylovora* decreased from 2.6 × 10^9^ CFU to 1.8 × 10^6^ CFU, while those of *P. carotovorum* subsp. *carotovorum* decreased from 4.7 × 10^9^ CFU to 1.2 × 10^8^ CFU. These findings suggest that *Herpetosiphon* strains exhibit a degree of prey preference.

In this study, highly susceptible pear leaves and *P. betulaefolia* seedlings (callery pear rootstock, used as a standard susceptible genotype) were employed as inoculation material, utilizing detached pear flower spray assays, detached pear leaf wound inoculation assays, detached pear fruit wound inoculation assays, and detached pear branch wound inoculation assays to accurately and objectively evaluate the biocontrol efficacy of *H. llansteffanensis* NSD29 against pear fire blight. The results demonstrated that inoculation with *H. llansteffanensis* NSD29 effectively controlled *E. amylovora* infection in pear leaves, young fruits, branches, and flowers, reducing disease severity. Recent studies have also reported promising biocontrol agents against fire blight. For instance, Zeller et al. [37] found that the *E. herbicola* strain 89, isolated from fruit tree endophytes, inhibited pear blossom blight by 70%. Bahadou et al. [38] evaluated ten bacterial strains (including *Alcaligenes*, *Pantoea*, *Serratia*, and *Bacillus*) for their ability to control fire blight on pear fruits, demonstrating significant disease reduction with varying levels of efficacy. Notably, several strains completely inhibited lesion expansion and necrosis on immature pear fruits, while others showed less than 50% control. Sharifazizi et al. [39] reported that *Pseudomonas* sp. Ps170, *Pseudomonas* sp. Ps117, *Enterobacter* sp. En113, and *P. agglomerans* Pa21 reduced leaf infection by 77%, 76%, 73%, and 71%, respectively, with *P. agglomerans* Pa21 showing the highest biocontrol activity (83% reduction) on immature fruits. Han et al. [40] demonstrated that *Myxococcus fulvus* WCH05 provided 76.02% protective efficacy against pear blossom blight at 5 dpi. Bai et al. [41] evaluated the protective effect of several myxobacteria against pear fire blight using detached leaves, showing that myxobacterial inoculation inhibited lesion expansion in leaf veins and reduced lesion length, indicating a certain level of biocontrol activity. The *Herpetosiphon* strain NSD29 isolated in this study exhibited comparable or superior control efficacy against *E. amylovora* compared with these recently reported antagonistic bacterial strains, highlighting its significant biocontrol potential and confirming *Herpetosiphon* as a promising source for highly active antimicrobial strains.

While this study has demonstrated the significant biocontrol effect of *H. llansteffanensis* NSD29 on fire blight under controlled conditions on pear flowers, leaves, young fruits, and detached branches, its persistence, colonization efficiency, and control efficacy in natural field environments particularly on whole pear plants require further investigation. One objective is to evaluate the biocontrol efficacy under field conditions. The other is to observe the plant’s multifaceted response, including potential growth promotion or adverse effects beyond disease control, as well as the dynamics of plant-associated microbiota. Furthermore, the potential for resistance emergence with long-term application of the biocontrol agent is a significant concern. Additionally, the severe aggregation of NSD29 into clusters during liquid culture may impede its application as a spray. Therefore, future research should focus on developing strategies to enhance cell dispersion during *Herpetosiphon* growth and to establish and optimize fermentation processes for its large-scale production.

## 4. Materials and Methods


### 4.1. Bacterial Strains and Culture Conditions


Five plant pathogenic bacterial strains were utilized in this study. *D. fangzhongdai* GMCC1.15464 and *P. carotovorum* subsp. *carotovorum* Eu678364 were kindly provided by Dr. Zhoukun Li (Nanjing Agricultural University). *A. citrulli* FC440 was generously donated by Dr. Jun Liu (Xinjiang Agricultural University). *P. syringae* pv. *syringae* ATCC 19310 was obtained from Professor Xianglin Zhang (Urumqi Customs). *E. amylovora* Ea001 was isolated and identified from *P. sinkiangensis* branches exhibiting fire blight symptoms in Korla, Xinjiang, China [42].

*E. coli* DH5α (Beijing Takara Biomedical Technology Co., Ltd., 9057, Beijing, China) served as the bait for *Herpetosiphon* isolation.

WCX agar (Water Calcium and Cycloheximide) medium (1 g/L CaCl_2_·2H_2_O, 15 g/L agar, 25 μg/mL cycloheximide, pH 7.2) [43] was used for isolating gliding filamentous bacteria. TPM (Tris Phosphate Magnet) medium (10 mM/L Tris-HCl, 1 mM/L KH_2_PO_4_, 8 mM/L MgSO_4_, 15 g/L agar, pH 7.6) [29] was employed for the in vitro predatory assays.

All five plant pathogenic strains were routinely cultured on Nutrient Agar (NA) (10 g/L peptone, 5 g/L NaCl, 3 g/L beef extract, pH 7.2) or in NA broth at 30 °C. The strain NSD29 was maintained on VY/2 agar (5 g/L yeast extract, 1 g/L CaCl_2_·2H_2_O, 15 g/L agar, pH 7.2) [44] or LBS broth (7 g/L soluble starch, 5 g/L yeast extract, 1 g/L casitone, 1 g/L MgSO_4_·7H_2_O, pH 7.2) [45].

### 4.2. Isolation and Purification of Herpetosiphon


In July 2021, natural forest soil samples were collected from the Tianshan Grand Canyon (42°09′–43°28′ N, 87°12′–87°50′ E, altitude 1846–2360 m) in Urumqi, Xinjiang Uygur Autonomous Region, China.

*H. llansteffanensis* NSD29 was isolated using a prey-induced method on WCX medium with *E. coli* DH5α as bait, following the protocol described by Yi et al. [46]. A window-shaped area was created on the WCX agar plate by applying an *E. coli* DH5α suspension (OD_600_ = 1.0). Cycloheximide-pretreated soil samples (0.7 g) were then placed in the center of each quadrant (Figure 1A). Plates were incubated at 30 °C for 72 h.

Under a stereomicroscope, fruiting body-like structures resembling myxobacteria were selected using a sterile fine needle and transferred to VY/2 agar for further cultivation at 30 °C. Following the isolation of pure cultures, bacterial films were scraped and suspended in 20% glycerol for long-term storage at −80 °C.

### 4.3. Morphological Observations


The colony morphology of the strain NSD29 was examined on VY/2 agar plates. The cellular morphology of NSD29 was characterized using scanning electron microscopy (SEM, SUPRA55VP, Zeiss, Jena, Germany) and transmission electron microscopy (TEM, HT7800, HITACHI, Tokyo, Japan). SEM observations were conducted at the Xinjiang Institute of Ecology and Geography, Chinese Academy of Sciences, and TEM observations were performed at Wuhan Seville Biotechnology Co., Ltd. (Wuhan, China). Briefly, bacterial samples for SEM were prepared on a dialysis bag (8–14 kDa molecular weight cutoff) placed on a VY/2 plate. Samples were fixed in FAA for 16 h, followed by dehydration through a graded ethanol series and subsequent soaking in tert-butanol. For TEM, bacteria were cultured in LBS for 72 h to an optical density at 600 nm (OD_600_) of 0.4. The bacterial pellet was then embedded in agarose and post-fixed with 1% osmium tetroxide (OsO_4_) in 0.1 M phosphate buffer (PB, pH 7.4) for 2 h. Ultrathin sections were cut, stained with saturated 2% uranyl acetate and 2.6% lead citrate, and then air-dried prior to observation.

### 4.4. Biochemical Characterization


Biochemical characterization of the strain NSD29 was performed using a Microbiochemical Identification Kit (Haibo Biotechnology Co., Ltd., Qingdao, China) following the manufacturer’s instructions. The resulting biochemical profile was compared with published data for other *Herpetosiphon* strains [25,29].

### 4.5. 16S rRNA Gene Analysis

For genomic DNA extraction, the strain NSD29 was cultured in 3 mL of LBS broth at 30 °C with shaking at 180 rpm for 48 h. Cells were harvested by centrifugation at 12,000 rpm for 1 min. Total DNA was extracted using the TIANamp Bacteria DNA Kit (TIANGEN Biotech, Beijing, China) according to the manufacturer’s protocol. The 16S rRNA gene was amplified using the universal primers 27F and 1492R [47]. The resulting PCR product was purified and sequenced using the Sanger method (Sangon Biotech Co., Shanghai, China). The obtained 16S rRNA gene sequence was subjected to BLAST analysis against the NCBI database. Phylogenetic analysis was conducted using MEGA7.0 software [48]. The 16S rRNA gene sequences were aligned using the ClustalW algorithm, and a phylogenetic tree was constructed using the neighbor-joining method with 1000 bootstrap replicates to assess nodal support.

### 4.6. Genome Sequencing and Analysis


Whole-genome sequencing of the strain NSD29 was performed using the Illumina NovaSeq platform (Nanjing Personal Gene Technology Co., Ltd., Nanjing, China) with 2 × 150 bp paired-end sequencing. Raw reads were trimmed to remove sequencing adapters using A5-MiSeq (v20160825) [49]. Genome assembly was performed using SPAdes (v3.15.4) [50]. The assembled genome was subjected to base correction using Pilon (v1.24) [51]. Protein-coding genes were predicted using GeneMarkS (v4.32) software [52] and functionally annotated using EggNOG-mapper [53]. Transfer RNAs (tRNAs), ribosomal RNAs (rRNAs), and other non-coding RNAs were predicted using tRNAscan-SE (v2.0) [54], Barrnap (v0.9), and the Rfam database (v14.1) [55].

Average nucleotide identity (ANI) and digital DNA–DNA hybridization (dDDH) values were calculated for species delineation [26,56] between the genome sequence of the strain NSD29 and the reference genomes of *Herpetosiphon* type strains using the online tools available at ANI: https://www.ezbiocloud.net/tools/ani (accessed on 10 May 2025); dDDH: https://ggdc.dsmz.de/ggdc_background.php (accessed on 10 May 2025).

The AntiSMASH web server (https://antismash.secondarymetabolites.org/) (accessed on 10 May 2025) was utilized to analyze the genome for secondary metabolite biosynthetic gene clusters [57]. The circular genome map of NSD29 was generated using the CGView Server [58].

### 4.7. Predatory Activity Against Plant Pathogens


The predatory activity of the strain NSD29 against the five plant pathogenic bacteria described in Section 4.1 was evaluated using a plate-based assay. Prey target bacterial suspensions (30 µL, OD_600_ = 1.0) were spotted onto TPM agar plates to create prey lawns and allowed to air-dry for 30 min. Subsequently, a 2 µL drop of the strain NSD29 suspension (OD_600_ = 1.0) was placed at the center of each prey lawn. The plates were incubated at 30 °C for 5 days, and the expansion of the NSD29 colony was visually monitored to assess predation.

To quantify predatory efficiency, the bacterial lawn was carefully scraped from the agar plate using a sterile loop and resuspended in 1 mL of sterile distilled water. Serial dilutions were prepared, and the number of viable colony-forming units (CFUs) of each pathogenic bacterium was determined by plating onto NA medium. The reduction in viable prey cell counts was calculated to evaluate the predatory capacity of the strain NSD29.

The predatory process was documented visually using an SM7 Motic microscope (McAudi Industrial Group Co., Ltd., Xiamen, China).

### 4.8. Biological Assay of the Strain NSD29 Against Pear Fire Blight


#### 4.8.1. Bacterial Preparation


The strain NSD29 was cultured in 300 mL of LBS broth in a 500 mL Erlenmeyer flask containing 200 glass beads (3 mm diameter) to promote cell dispersion. The culture was incubated at 30 °C with shaking at 180 rpm for 3 days. The resulting bacterial suspension was thoroughly mixed to obtain the NSD29 inoculum (OD_600_ = 1.0). *E. amylovora* was cultured in LB broth and then diluted to a concentration of 1 × 10^7^ CFU/mL with sterile water.

#### 4.8.2. Assay on Fragrant Pear Inflorescences


Fragrant pear (*P. sinkiangensis*) inflorescences attached to approximately 20 cm long branches were collected from a pear orchard in Korla, Xinjiang (April 2022), and immediately placed in bottles containing a 3% sucrose solution. Four treatments were applied to fifty inflorescences per treatment, with three replicates, and the assay was repeated twice: (A) inoculated with *E. amylovora* (10 mL, 1 × 10^7^ CFU/mL) only; (B) pre-inoculated with the strain NSD29 (10 mL, OD_600_ = 1.0), followed by *E. amylovora* (10 mL, 1 × 10^7^ CFU/mL) 24 h later (protective treatment); (C) inoculated with a 3% benziothiazolinone WDG solution (BIT, Andama Hui Feng Co., Jiangsu, Dafeng, China) at an 800-fold dilution and then treated with *E. amylovora* (10 mL, 1 × 10^7^ CFU/mL); and (D) mock control: treated with sterile water only. The treated inflorescences were incubated in a growth chamber at 28 °C and 70% relative humidity for 5 days, and the incidence of blossom blight was recorded daily. Control efficacy was calculated based on the mock-treated control.

#### 4.8.3. Assay on Detached Fragrant Pear Leaves


Healthy, uniformly sized young leaves of fragrant pear were collected and rinsed with sterile water. After air-drying, a wound was created at the junction of the leaf blade and petiole using a sterile needle. Four treatments were applied to three leaves per treatment, with three replicates: (A) inoculated with *E. amylovora* (2 μL, 1 × 10^7^ CFU/mL) only; (B) pre-inoculated with the strain NSD29 (2 μL, OD_600_ = 1.0), followed by *E. amylovora* (2 μL, 1 × 10^7^ CFU/mL) 24 h later (protective treatment); (C) inoculated with a 3% benziothiazolinone WDG solution (BIT, 800-fold dilution) and then treated with *E. amylovora* (2 μL, 1 × 10^7^ CFU/mL); and (D) mock control: treated with sterile water only. The inoculated leaves were placed in sterile plastic boxes lined with moistened sterile filter paper and incubated at 28 °C in a growth chamber. Lesion length was measured daily for 5 days, and control efficacy was calculated using the following formula:Control Efficacy (%) = [(Lesion Length (Control) − Lesion Length (Treatment)]/Lesion Length (Control) × 100

#### 4.8.4. Assay on Detached Fragrant Pear Fruits


Uniformly sized, immature fragrant pear fruits were selected, rinsed with sterile water, and surface-sterilized with 75% ethanol for 5 min, followed by air-drying. Three small wounds (2 mm wide, 5 mm deep) were created on each fruit using a sterile pipette tip. Four treatments were applied to three fruits per treatment, with three replicates: (A) inoculated with *E. amylovora* (3 μL, 1 × 10^7^ CFU/mL) only (positive control); (B) pre-inoculated with the strain NSD29 (3 μL, OD_600_ = 1.0), followed by *E. amylovora* (3 μL, 1 × 10^7^ CFU/mL) 24 h later (protective treatment); (C) inoculated with a 3% benziothiazolinone WDG solution (BIT, 800-fold dilution) and then treated with *E. amylovora* (3 μL, 1 × 10^7^ CFU/mL); and (D) mock control: treated with sterile water only. The inoculated fruits were placed in sterile plastic boxes lined with moistened sterile filter paper and incubated at 28 °C in a growth chamber. The lesion diameter was measured daily for 5 days, and the disease index and control efficacy were calculated.

Disease severity was assessed using a modified scale based on Medhioub et al. [59]: 0 = no necrotic lesion; 1 = necrotic lesion diameter 1–4 mm; 2 = necrotic lesion diameter 5–8 mm; 3 = necrotic lesion diameter 9–12 mm; and 4 = necrotic lesion diameter 13–16 mm.Disease Index = ∑ (Number of Fruits in Each Disease Grade × Representative Value of Each Grade)/Total Number of Inoculated Fruits × Highest Disease Grade Value × 100Control Efficacy (%) = [(Disease Index (Positive Control) − Disease Index (Treat-ment)]/Disease Index (Positive Control) × 100

#### 4.8.5. Assay on Detached Fragrant Pear Branches


For protective efficacy assays, current-year fragrant pear shoots (approximately 30 cm in length) were collected, and five wounds were made 2 cm from the apical tip using a sterile dissecting needle. The wound had a depth of 4 mm and a width of 1 mm. Strain NSD29 suspension (OD_600_ = 1.0) or a 3% benziothiazolinone WDG solution (800-fold dilution) was sprayed evenly onto the shoots. After 24 h, 5 μL of *E. amylovora* suspension (1 × 10^7^ CFU/mL) was inoculated into each wound, covered with moistened sterile cotton, and wrapped with parafilm for 48 h. The cotton and parafilm were then removed, and the branches were incubated at 28 °C and 70% relative humidity in a growth chamber. Four treatments were applied to five branches per treatment, with three replicates: (A) inoculated with *E. amylovora* (5 μL, 1 × 10^7^ CFU/mL) only; (B) pre-inoculated with the strain NSD29 (OD_600_ = 1.0), followed by *Ea* (5 μL, 1 × 10^7^ CFU/mL) 24 h later (protective treatment); (C) inoculated with a 3% benziothiazolinone WDG solution (BIT, 800-fold dilution) and then treated with *E. amylovora* (5 μL, 1 × 10^7^ CFU/mL); and (D) mock control: treated with sterile water only.

For curative efficacy assays, the inoculation sequence of NSD29 and *E. amylovora* was reversed compared with the protective assay. First, 5 μL of *E. amylovora* suspension was inoculated into the wounds at the apical tip of the pear shoots. After 24 h, the shoots were sprayed with either NSD29 suspension or benziothiazolinone solution. The remaining procedures were identical to the protective assay. Four treatments were applied to five branches per treatment, with three replicates: (A) inoculated with *E. amylovora* (5 μL, 1 × 10^7^ CFU/mL) only; (B) inoculated with *E. amylovora* (5 μL, 1 × 10^7^ CFU/mL), followed by the strain NSD29 (OD_600_ = 1.0) 24 h later (curative treatment); (C) inoculated with *E. amylovora* (5 μL, 1 × 10^7^ CFU/mL) and then treated with a 3% benziothiazolinone WDG solution (BIT, 800-fold dilution); and (D) mock control: treated with sterile water only. Disease development was monitored for 7 days, and lesion length was measured to calculate control efficacy using the following formula:Control Efficacy (%) = [(Lesion Length (Control) − Lesion Length (Treatment)]/Lesion Length (Control) × 100

### 4.9. Statistical Analysis


Statistical analyses were performed using one-way analysis of variance (ANOVA), followed by Duncan’s multiple range test to determine significant differences between treatments. ANOVA was conducted using SPSS 19.0 software (IBM Corp., Armonk, NY, USA). A significance level of *p* < 0.05 was considered statistically significant. Data analysis and figure generation were performed using GraphPad Prism 8 software (GraphPad Software, San Diego, CA, USA).

## 5. Conclusions


This study has successfully isolated and characterized a novel bacterial strain, *H. llansteffanensis* NSD29 (SRA accession number PRJNA1255817), from forest soil in Xinjiang, China, as a promising candidate for sustainable pear fire blight management. The comprehensive in vitro analyses demonstrate that NSD29 exhibits significant predatory activity against a range of plant pathogenic bacteria, including *E. amylovora*, the causal agent of fire blight. Furthermore, in planta bioassays conducted on detached pear blossoms, leaves, fruits, and branches consistently revealed the biocontrol efficacy of the strain NSD29 against *E. amylovora*, comparable to the chemical fungicide benziothiazolinone. These findings strongly suggest that *H. llansteffanensis* NSD29 represents a novel and effective biocontrol agent for pear fire blight, offering an environmentally sound alternative to conventional chemical treatments. The unique predatory lifestyle of this strain, coupled with its demonstrated ability to suppress fire blight symptoms across various pear tissues, underscores the potential of *Herpetosiphon* species as a valuable resource for developing innovative and sustainable disease management strategies in agriculture.

## Figures and Tables

**Figure 1 plants-14-01564-f001:**
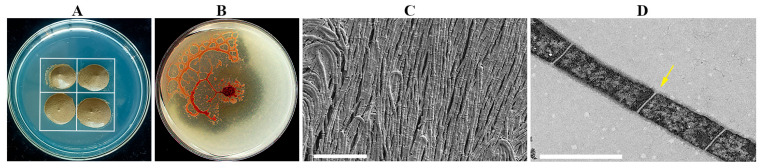
Morphological characterization of the strain NSD29. (**A**) Schematic diagram illustrating the prey-induced isolation method. The white outline delineates the *Escherichia coli* bacterial lawn. (**B**) Colony morphology of the strain NSD29 cultivated on VY/2 agar (5 d). (**C**) Scanning electron micrograph (SEM) of strain NSD29 filaments. Scale bar = 10 µm. (**D**) Transmission electron micrograph (TEM) of the strain NSD29, revealing chained cells with constrictions (arrow) between adjacent cells. Scale bar = 1 µm.

**Figure 2 plants-14-01564-f002:**
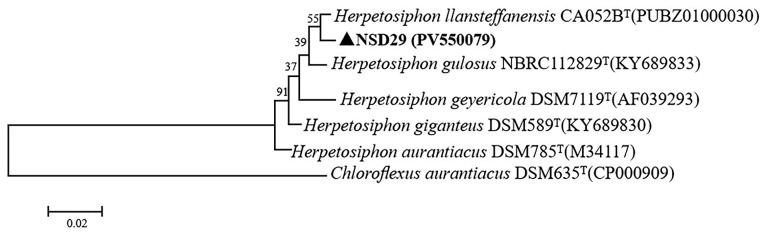
Phylogenetic analysis of the strain NSD29 based on 16S rRNA gene sequences. *Chloroflexus aurantiacus* DSM 635ᵀ (CP000909), a member of the class Chloroflexota, was used as the outgroup.

**Figure 3 plants-14-01564-f003:**
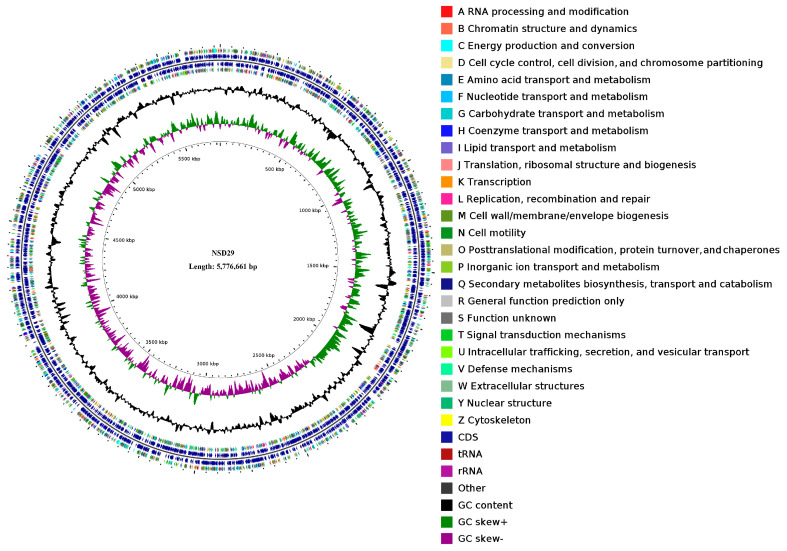
Circular genome map of the strain NSD29. From the innermost circle outwards: (1) scale, (2) GC skew, (3) GC content, (4 and 7) COG categories of CDS, (5 and 6) genomic location of CDS, tRNA, and rRNA genes.

**Figure 4 plants-14-01564-f004:**
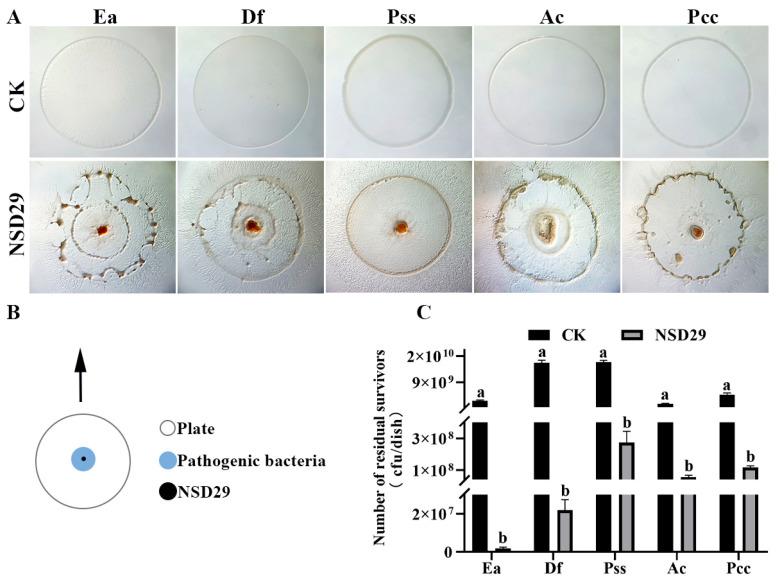
Predatory activity of the strain NSD29 against multiple plant pathogenic bacteria. (**A**) Predation assay setup: 30 µL suspensions of target prey bacteria (OD_600_ = 1.0) were spotted onto TPM agar to create prey lawns and allowed to dry for 30 min. Subsequently, a 2 µL drop of strain NSD29 suspension (OD_600_ = 1.0) was placed at the center of each prey lawn. (**B**) Schematic representation of the predation process. (**C**) Quantification of predation efficiency, determined by scraping the bacterial lawn, serially diluting, and plating to enumerate viable pathogenic bacteria. For panels (**A**,**C**), “CK” denotes the control group inoculated only with the pathogen. Ea: *Erwina amylovora*; Df: *Dickeya fangzhongdai*; Pss: *P. syringae* pv. *Syringae*; Ac: *Acidovorax citrulli*; and Pcc: *Pectobacterium carotovorum* subsp. *carotovorum.* Error bars represent the standard error of the mean (*n* = 3). Statistical significance was assessed using Duncan’s multiple range test (*p* < 0.05). Treatments sharing the same letter are not significantly different.

**Figure 5 plants-14-01564-f005:**
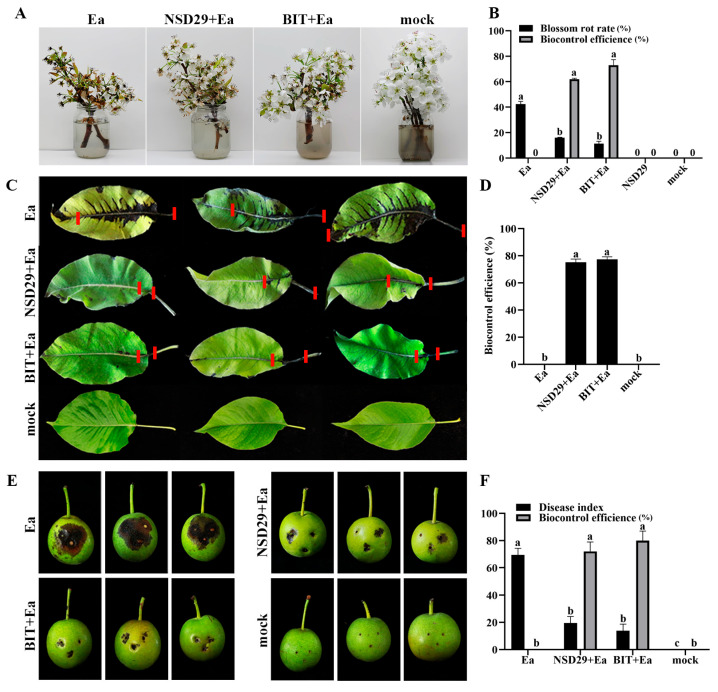
Bioassay of NSD29 in controlling fire blight on pear blossoms, leaves, and fruits. (**A**) Control of fire blight on detached pear blossoms by the tested strain NSD29 (5 dpi). NSD29/BIT + Ea represents the protective treatment, where NSD29 and BIT were inoculated 24 h before Ea; mock: sterile water; Ea: inoculation with *E. amylovora* only; NSD29: inoculation with the strain NSD29 only; and BIT: inoculation with benziothiazolinone. (**B**) Biological control effect of the strain NSD29 on fire blight of pear blossoms (5 dpi). (**C**) Control of fire blight on detached pear leaves by the tested strain NSD29 (5 dpi). On the same leaf, the two short red lines define the expanded distance of the disease symptom along the vein. (**D**) Biological control effect of the strain NSD29 on fire blight of pear leaves (5 dpi). (**E**) Control of fire blight on detached pear fruits by the tested strain NSD29 (5 dpi). (**F**) Biological control effect of the strain NSD29 on fire blight of pear fruits (5 dpi). NSD29/BIT + Ea represents the protective treatment, where NSD29 and BIT were inoculated 24 h before Ea; mock: sterile water; Ea: inoculation with *E. amylovora* only; and BIT: inoculation with benziothiazolinone. Different lowercase letters (a, b, c) in panels (**B**,**D**,**F**) indicate a significant difference at *p* = 0.05.

**Figure 6 plants-14-01564-f006:**
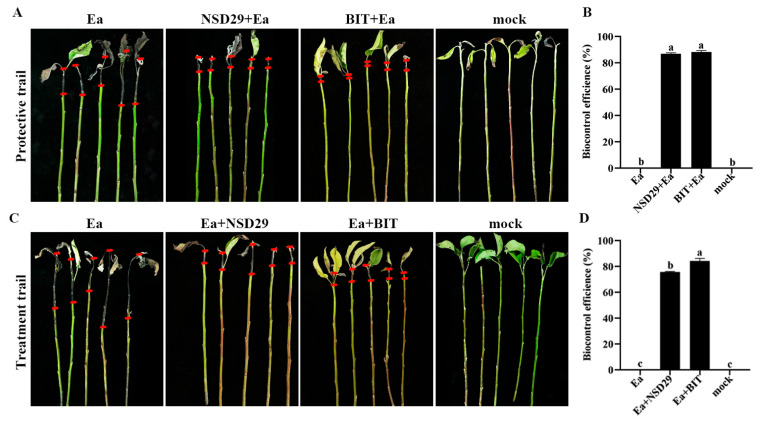
Biocontrol efficacy of the strain NSD29 against fire blight on detached pear branches. (**A**) Representative images of the protective assay on pear branches 7 days after inoculation (dpi). (**B**) Quantification of the protective biocontrol efficacy on detached pear branches. (**C**) Representative images of the curative assay on pear branches 7 dpi. (**D**) Quantification of the curative biocontrol efficacy on detached pear branches. Protective treatment: NSD29 + *Ea* and BIT + *Ea* (NSD29 or benziothiazolinone applied 24 h before *E. amylovora*). Curative treatment: *Ea* + NSD29 and *Ea* + BIT (*E. amylovora* inoculated 24 h before NSD29 or benziothiazolinone). Control: *Ea* (pathogen only). On the same shoot, the two short red lines define the expanded distance of the disease symptom from the inoculation site. Different lowercase letters (a, b, c) in panels (**B**,**D**) indicate a significant difference at *p* = 0.05.

**Table 1 plants-14-01564-t001:** Ability of the strain NSD29 and five other strains to metabolize different biochemical substrates.

Biochemical Substrates	*H. llansteffanensis* CA052B^T^	*H. aurantiacus* DSM785^T^	*H. geysericola* DSM7119^T^	*H. gulosus* NBRC 112829^T^	*H. giganteus* DSM589^T^	NSD29
Esculin	+	−	+	+	+	+
Arginine	−	+	−	+	+	+
Gelatin	+	−	−	−	+	+
Glucose	+	+	−	+/−	+/−	−
Arabinose	−	+	+	+	+	−
Sucrose	−	+	+	+	+	+
Rhamnose	−	+	+	+	+	−
Mannitol	−	+	+	+	+	−

Note: +, positive; +/−, weakly positive; −, negative. Data for strains other than NSD29 are from Livingstone et al.

**Table 2 plants-14-01564-t002:** Genomic features of sequenced *Herpetosiphon* spp.

Characteristic	*H. llansteffanensis* CA052B^T^	*H. aurantiacus* DSM785^T^	*H. geysericola* DSM7119^T^	NSD29
Size (Mbp)	6.14	6.79	6.24	5.78
No. of contigs	170	1 (+2 plasmids)	46	1 (+4 plasmids)
%GC content	50.8	50.9	50.7	50.89
t1PKS-NRPS	2	3	2	2
Thiopeptide	1	1	1	1
Terpene	2	2	2	2
Bacteriocin	2	1	2	−
NRPS	2	4	2	2
t3PKS	1	1	1	1
t1PKS	−	1	−	−
Lantipeptide-t1PKS-NRPS	−	1	−	−

Note: Data of *H. llansteffanensis* CA052B^T^, *H. aurantiacus* DSM 785^T^, and *H. geysericola* DSM 7119^T^ from Livingstone [25].

**Table 3 plants-14-01564-t003:** Number of genes associated with the general COG functional categories.

Code	Value	%	Description
B	2	0.05	Chromatin structure and dynamics
C	277	6.87	Energy production and conversion
D	41	1.02	Cell cycle control, cell division, and chromosome partitioning
E	255	6.32	Amino acid transport and metabolism
F	131	3.25	Nucleotide transport and metabolism
G	256	6.35	Carbohydrate transport and metabolism
H	192	4.76	Coenzyme transport and metabolism
I	128	3.17	Lipid transport and metabolism
J	186	4.61	Translation, ribosomal structure, and biogenesis
K	400	9.92	Transcription
L	199	4.93	Replication, recombination, and repair
M	275	6.82	Cell wall/membrane/envelope biogenesis
N	33	0.82	Cell motility
O	123	3.05	Post-translational modification, protein turnover, and chaperones
P	208	5.16	Inorganic ion transport and metabolism
Q	110	2.73	Secondary metabolites biosynthesis, transport, and catabolism
S	707	17.53	Function unknown
T	333	8.26	Signal transduction mechanisms
U	85	2.11	Intracellular trafficking, secretion, and vesicular transport
V	88	2.18	Defense mechanisms
W	2	0.05	Extracellular structures
Z	2	0.05	Cytoskeleton

**Table 4 plants-14-01564-t004:** Comparative analysis of the strain NSD29 and five closely related reference strains based on 16S rRNA sequence identity, average nucleotide identity (ANI), and digital DNA–DNA hybridization (dDDH) values.

Strains (GenBank Accession Number)	16S rRNA Identity (%)	ANI (%)	dDDH (%)
*H. llansteffanensis* CA052B^T^ (PUBZ01000117)	99.2	96.1	66.2
*H. giganteus* DSM589^T^ (NZ_JAFBCZ010000001)	99.2	91.7	45.3
*H. gulosus* NBRC112829^T^ (BAABRU010000059)	99.2	84.5	28.1
*H. geysericola* DSM7119^T^ (LGKP01000036)	98.2	91.1	43.2
*H. aurantiacus* DSM785^T^ (CP000875)	95.5	84.4	28.2

## Data Availability

The genome sequence was deposited in NCBI with accession number PV550079, NSD29 GenBank accession number CP188824, and SRA accession number PRJNA1255817. Other original data will be available upon request from the corresponding author.

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
