# Peer review of "A Herpetosiphon llansteffanensis Strain from Forest Soil Exhibits Biocontrol Activity Against Pear Fire Blight"

_plants, 2025, doi:10.3390/plants14111564_

Round 1

Reviewer 1 Report

Comments and Suggestions for Authors

This is a very valuable manuscript. It is devoted to the serious disease, pear fire blight, caused by the bacterium Erwinia amylovora. The methodological aspects are well presented. The authors isolated a bacterial strain NSD29 from forest natural soil, which displayed antagonistic activity against five Gram-negative plant pathogenic bacteria. Then, the bacterium was characterized, taking into account morphology, physiological and biochemical aspects, 16S rRNA gene sequencing, and genomic features. Based on this integrated analysis, it was concluded that strain NSD29 represents a novel strain within the species Herpetosiphon llansteffanensis. Bioassays conducted under green house condition on detached pear blossoms, leaves, fruits, and branches revealed the potent biocontrol efficacy of strain H. llansteffanensis NSD29 against E. amylovora. These results were compared to the results of the use of the chemical fungicide benziothiazolinone. All results are well documented and clearly presented in tables and figures. They indicate that H. llansteffanensis strain NSD29 may represent a new and effective biocontrol agent for pear fire blight which may be a good alternative to chemical fungicides. The manuscript is written carefully, the Discussion section is interesting. The manuscript should be published in Plants after minor revision. The comments in Remarks should be taken into account.

Remarks

Line 20-21 here the name Haliangium llansteffanensis is given. This requires explanation as it is probably an error

Line 23 it should be ‘in vivo’ instead of ‘ex vivo’ or cross it out as the experimental conditions were clearly defined (under controlled green house conditions)

Line 70 it should be ‘Pseudomonas’ instead of ‘P.’

Line 78 (and other places) according to current taxonomy it should be ‘Chloroflexota’ instead of ‘Chloroflexi’

Line 99 Fig - it should be Figure as in other places

Line 113 Figure 1B – the age of the colony should be given

Line 124 H.giganteus - for all names add a space after the dot (the same applies to Table 3)

Line 141 it should be Livingstone et al.

Line 172 Table 2 - or Table 3?

Line 207 Figure 4A and 4C - explain CK (probably control)

Line 254- 255 consider revising this sentence

Line 499 ‘five wounds were made’ - describe these wounds in more detail

Line 591 perforans - it should be in italic

Author Response

Dear reviewer,

We appreciate your meticulous work on this manuscript. The following is our response point-by-point.

Remarks

  1. Line 20-21 here the name Haliangium llansteffanensis is given. This requires explanation as it is probably an error

Response: We appreciate your meticulous reviewing and proofreading. There was an error, which we have corrected to Herpetosiphon llansteffanensis. We have also checked the entire manuscript for similar errors.

  1. Line 23 it should be ‘in vivo’ instead of ‘ex vivo’ or cross it out as the experimental conditions were clearly defined (under controlled green house conditions)

Response: As you suggested, we have deleted "in vivo".

  1. Line 70 it should be ‘Pseudomonas’ instead of ‘P.’

Response: We have revised it and checked the entire manuscript to ensure consistency with the general rule.

  1. Line 78 (and other places) according to current taxonomy it should be ‘Chloroflexota’ instead of ‘Chloroflexi’

Response: Thank you for your rigorous review. We have revised it accordingly.

  1. Line 99 Fig - it should be Figure as in other places

Response: We have revised all instances to use "Fig." instead of "Figure."

  1. Line 113 Figure 1B – the age of the colony should be given

Response: The culture time has been added to the end of the description.

  1. Line 124 giganteus - for all names add a space after the dot (the same applies to Table 3)

Response: Thank you. We have revised that and checked all Latin names in the manuscript to ensure they are italicized.

  1. Line 141 it should be Livingstone et al.

Response: We have deleted reference [26] following Livingstone et al.

  1. Line 172 Table 2 - or Table 3?

Response: We have checked all tables and figures to ensure they are presented in order. The table you mentioned is now Table 4.

  1. Line 207 Figure 4A and 4C - explain CK (probably control)

Response: We have added the description of CK in that experiment to the figure legend.

  1. Line 254- 255 consider revising this sentence (We isolated and characterized a gliding filamentous bacterium, llansteffanensis. NSD29, which presents a promising addition to the disease control arsenal.)

Response: We have revised the sentence as shown in the new version.

  1. Line 499 ‘five wounds were made’ - describe these wounds in more detail

Response: We have provided the details of those wounds on the shoots.

Line 591 perforans - it should be in italic

Response: We have revised it and checked all Latin names to ensure they are italicized throughout the manuscript.

Besides, we have checked and revised other possible errors in the entire manuscript based on the suggestions and comments from both anonymous reviewers and the editor.

Reviewer 2 Report

Comments and Suggestions for Authors

An annotated revised version is provided. The main concerns to be addressed further in discussion are in antibiotics are involved in the action of the bacterium the concern for the molecules selecting strain and others is still present and this must be underlined and stressed. Second and more important for this research, whole pear plants must be used to cross confirm the data on parts of plants since the experiment performed do not allow the presence of responses of the plant to the presence of microorganisms so the result could be diverse from those presented here. Both issues should be further discussed and clarified in the revised version.

Author Response

Dear reviewer,

We appreciate your time and effort in inputting our manuscript. The following is the response point-by-point raised by you. 

  1. The main concerns to be addressed further in discussion are in antibiotics are involved in the action of the bacterium the concern for the molecules selecting strain and others is still present and this must be underlined and stressed.

Response: You are exactly right. Antibiotics application may cause resistance emerging among pathogens, the newly found biocontrol microorganisms/agents may weaken their efficacy due to pathogens’ resistance evolution. Regarding this concern, we add a sentence in the last paragraph of the discussion section. You are exactly right. Besides antibiotics application, the application of a microbial agent may lead to the emergence of resistance in pathogens as well, potentially weakening the efficacy of newly discovered biocontrol microorganisms/agents due to the evolution of pathogen resistance. To address this concern, we have added a sentence to the last paragraph of the discussion section.

  1. Second and more important for this research, whole pear plants must be used to cross confirm the data on parts of plants since the experiment performed do not allow the presence of responses of the plant to the presence of microorganisms so the result could be diverse from those presented here. Both issues should be further discussed and clarified in the revised version.

Response: We appreciate your constructive comments. This refers to our planned field test with whole plants, which may last for at least two years. Therefore, no data from this experiment is included in the current paper, and this point has been added to the last paragraph.

  1. PDF annotated revision

Response: We are very grateful for your meticulous revisions directly on the PDF! We have revised all points according to your suggestions in the Word version and added details where necessary, especially in the SEM and TEM methods sections. The Erwinia amylovora resource is already presented in the manuscript. All revisions are shown in track changes in the uploaded file.

Besides, we have checked and revised other possible errors in the entire manuscript based on the suggestions and comments from both anonymous reviewers and the editor.

Round 2

Reviewer 2 Report

Comments and Suggestions for Authors

The manuscript still has some typos to be revised, below the list:

Figure 1 the figure D is still not focus a better more focused picture must be added.

Lane 279 pv Syringae change to pv. syringae (in Italic)

Lane 330 E. amylovora must be in Italic

Lane 382 H. llansteffanensis mut be in Italic

Lane 478 erase in the future.

Lane 478-479 replace realistic with field

Author Response

Dear reviewer/editor,

We appreciate your meticulous work on this manuscript, which is crucial for maintaining the journal's high standards. Our point-by-point response follows.

We are confident that the manuscript has been significantly improved as a result of these revisions.

Figure 1 the figure D is still not focus a better more focused picture must be added.

Response: We have replaced Figure 1D with a higher-quality version that has improved focus.

Lane 279 pv Syringae change to pv. syringae (in Italic)

Response: Thank you for your careful checking. We have revised this error.

Lane 330 E. amylovora must be in Italic

Response: We have revised this.

Lane 382 H. llansteffanensis mut be in Italic

Response: We have revised this.

Lane 478 erase in the future.

Response: We have deleted  “in the future”.

Lane 478-479 replace realistic with field

Response: We have changed realistic to field.

Best regards,

Benzhong on behalf of all co-authors